# A New Gnotobiotic Pig Model of P[6] Human Rotavirus Infection and Disease for Preclinical Evaluation of Rotavirus Vaccines

**DOI:** 10.3390/v14122803

**Published:** 2022-12-15

**Authors:** Charlotte Nyblade, Casey Hensley, Viviana Parreño, Peng Zhou, Maggie Frazier, Annie Frazier, Ashwin Ramesh, Shaohua Lei, Juan Ignacio Degiuseppe, Ming Tan, Lijuan Yuan

**Affiliations:** 1Department of Biomedical Sciences and Pathobiology, Virginia-Maryland College of Veterinary Medicine, Virginia Tech, Blacksburg, VA 24060, USA; 2Incuinta, IVIT, INTA-Conicet, Hurlingham, Provincia de Buenos Aires 1686, Argentina; 3Laboratory of Viral Gastroenteritis, National Institute for Infectious Diseases (INEI-ANLIS “Dr. Carlos G. Malbrán”), Ciudad Autónoma de Buenos Aires 1281, Argentina; 4Division of Infectious Diseases, Cincinnati Children’s Hospital Medical Center, Cincinnati, OH 45229, USA; 5Department of Pediatrics, University of Cincinnati College of Medicine, Cincinnati, OH 45229, USA

**Keywords:** human rotavirus, P[6] genotype, gnotobiotic pig model, diarrhea, vaccine evaluation

## Abstract

Human rotavirus (HRV) is a leading cause of gastroenteritis in children under 5 years of age. Licensed vaccines containing G1P[8] and G1-4P[8] strains are less efficacious against newly emerging P[6] strains, indicating an urgent need for better cross protective vaccines. Here, we report our development of a new gnotobiotic (Gn) pig model of P[6] HRV infection and disease as a tool for evaluating potential vaccine candidates. The Arg HRV (G4P[6]) strain was derived from a diarrheic human infant stool sample and determined to be free of other viruses by metagenomic sequencing. Neonatal Gn pigs were orally inoculated with the stool suspension containing 5.6 × 10^5^ fluorescent focus units (FFU) of the virus. Small and large intestinal contents were collected at post inoculation day 2 or 3. The virus was passaged 6 times in neonatal Gn pigs to generate a large inoculum pool. Next, 33–34 day old Gn pigs were orally inoculated with 10^−2^, 10^3^, 10^4^, and 10^5^ FFU of Arg HRV to determine the optimal challenge dose. All pigs developed clinical signs of infection, regardless of the inoculum dose. The optimal challenge dose was determined to be 10^5^ FFU. This new Gn pig model is ready to be used to assess the protective efficacy of candidate monovalent and multivalent vaccines against P[6] HRV.

## 1. Introduction

Group A human rotaviruses (HRV) remain a leading cause of diarrheal disease in children under the age of five [1]. Vaccination has been an important strategy for mitigating the burden of disease. The two most widely implemented vaccines, Rotarix and RotaTeq, containing the viral strains G1P[8] and G1-4P[8], respectively, show high efficacy against the predominantly circulating strains of HRV [2,3,4]. However, these vaccines are much less efficacious against P[6] strains of HRV that have been emerging across the globe [4]. For comparison, one study found Rotarix to be ~55% efficacious against P[6] strains versus ~60% efficacy against P[8] and 70% efficacy to P[4] [4]. G4P[6] strains have been isolated from children in several countries across the globe, including Argentina, Sri Lanka, Italy, and Korea [5,6,7,8]. A G10P[6] strain was also isolated in addition to G4P[6] from children in northeastern India [9]. Interestingly, genomic analysis of the G4P[6] strains suggested that they are of porcine origin [5,6,7,9]. Pigs are highly susceptible to many rotavirus genogroups, including the human predominant Group A HRVs [10]. The virus spreads easily in herds and infected animals may or may not develop diarrhea [11].

Although rotaviruses typically infect hosts in a species-specific manner, multiple studies have shown that zoonotic transmission and genomic reassortment can occur [10]. Analysis of unusual strains across the globe using genomic sequencing or RNA/RNA hybridization techniques indicated that these strains contain genes of porcine origin [10]. Furthermore, the low prevalence of uncommon strains (20%) compared to the most commonly circulating strains of rotavirus (63%) along with their limited person-to-person transmission ability implies that they are a result of interspecies transmission and reassortment [5,8,9,12]. These data highlight swine as important source of reassorted rotaviruses.

With the rise of G4P[6] strains with zoonotic potential, a concerted effort needs to be made to tailor vaccines to provide cross protection against P[8] and P[6] strains. The development of vaccines is aided by the availability of an animal model that mimics the infection route, pathogenesis, and resolution of the disease. There is currently no animal model available to evaluate the protective efficacy against P[6] HRV infection and disease in preclinical vaccine trials. In the past, studies of rotavirus have been supported by the gnotobiotic (Gn) pig model of virulent Wa (G1P1A[P8]) HRV infection and disease. This model was established by serial passaging of Wa HRV through neonatal Gn pigs to generate an inoculum pool of infected intestinal contents [13]. This pool was then used to infect 4–5 day old Gn pigs and determine the median infectious dose (ID_50_), median diarrhea dose (DD_50_), and morphological changes in the intestinal tract caused by the infection [13]. All of the animals shed viruses and developed small intestinal lesions localized to the jejunum and ileum [13]. The consistencies in clinical presentation to human children in addition to similar immune system physiology and development make Gn pigs invaluable for modeling HRV infection and disease. The goal of this study was to establish a new Gn pig model of P[6] HRV infection and disease.

## 2. Materials and Methods

### 2.1. Virus Passaging in Neonatal Gn Pigs

Gn pigs were derived via hysterectomy and housed in sterile isolators for the study duration as previously described [14]. All pigs were fed ultra-high temperature sterilized whole cow’s milk (The Hershey Company, Hershey, PA, USA) and remained bacteria-free throughout the study as confirmed by weekly rectal swabs (RS) tested on blood agar plates and thioglycolate broth. All gnotobiotic pig study protocols were approved by the Institutional Animal Care and Use Committee at Virginia Polytechnic Institute and State University (CVM-19-235, 12/16/2019).

To generate the inoculum, an infant stool sample (Arg 12461) was centrifuged at 1026× *g* for 15 min at 4 °C. The supernatant was transferred to 1.5 mL Eppendorf tubes and centrifuged again at 20,000× *g* for 30 min to pellet any cells and bacteria. The supernatant was then collected and added to sterile PBS with 1% penicillin/streptomycin. Sterility was tested by culturing 10 µL virus on blood agar plates (Hardy Diagnostics A10, Santa Maria, CA, USA) and in Thioglycolate media (Hardy Diagnostics C7501, Santa Maria, CA, USA) for 3 days. Pigs received 4 mL of 200 mM sodium bicarbonate 10 min prior to inoculation to neutralize stomach acidity. At passage 0 (P0), pigs were orally inoculated with 100 µL of the infant stool specimen (diluted in 5 mL of Diluent #5 [minimal essential medium with 1% penicillin-streptomycin and 1% HEPES] containing 5.6 × 10^6^ fluorescent focus units (FFU)/mL of Arg G4P[6] HRV (from Degiuseppe, Instituto Nacional de Enfermedades Infecciosas, Buenos Aires, Argentina). At necropsy on post-inoculation day (PID) 2 or 3, intestinal contents were collected and used to continue serially passaging of the virus in Gn pigs.

### 2.2. Next Generation Sequencing

Next generation sequencing was performed to confirm the absence of other RNA or DNA viruses. To extract viral RNA, 200 µL of fecal samples were mixed with 600 µL of Trizol LS (Invitrogen 10296-010, Waltham, MA, USA), vortexed for 30 s, and incubated for 5 min at room temperature. 160 µL of 100% chloroform was added, then the sample was vortexed, incubated for 10 min at room temperature, and centrifuged for 18,407× *g* for 15 min at 4 °C. The supernatant was removed and mixed with 400 µL of 100% isopropyl alcohol, incubated for 10 min at room temperature, and centrifuged at 18,407× *g* for 10 min at 4 °C. The supernatant was removed and mixed with 800 µL of 75% ethanol and centrifuged at 6010× *g* for 5 min at 4 °C. The supernatant was discarded and the pellet was allowed to air dry before resuspension in 40 µL of nuclease free water. The mixture was incubated in a 55 °C water bath for 10 min. DNA was extracted from fecal samples using the ZymoBIOMICS™ DNA Miniprep Kit (Zymo Research D4300, Irvine, CA, USA) according to the manufacturer’s instructions. Samples were submitted to the Genomics Sequencing Center at Virginia Polytechnic Institute and State University for analysis. RNA quality was evaluated with Agilent BioAnalyzer before next generation sequencing with Illumina NextSeq 1000.

### 2.3. Pathogenesis of Arg HRV

To characterize intestinal histopathological changes caused by Arg HRV infection, neonatal Gn pigs were orally inoculated with 10^5^ FFU of Arg HRV and euthanized at PID 3. The virus used for inoculation was from the 6th passage through neonatal Gn pigs. Intestines were evaluated in situ before being removed from the body cavity and examined for edema, hemorrhage, and other signs of inflammation. Sections of the duodenum, jejunum, and ileum were collected, fixed in 10% buffered formalin, and submitted to Virginia Tech Animal Laboratory Services (ViTALS) at Virginia-Maryland College of Veterinary Medicine for hematoxylin and eosin (H&E) staining and examination of histopathological changes. A board-certified veterinary pathologist examined the slides and provided the report.

### 2.4. Dose Response Study of Arg HRV in Gn Pigs

A dose response study was performed to determine the level of infectivity and an optimal challenge dose of Arg HRV in Gn pigs of 33 days of age. Arg HRV inoculum was adjusted to one of four doses (10^−2^, 10^3^, 10^4^, or 10^5^ FFU) in 5 mL of Diluent #5. Dose selection was based on previous studies evaluating the ID_50_ and DD_50_ of virulent Wa HRV in Gn pigs [13]. Pigs were orally inoculated with their respective dose at 33 days of age and monitored for 7 days (PID 0–7). RS were collected daily from PID 0–7 and used to evaluate fecal consistency and virus shedding. Fecal consistency was based on a score of 0–3 (0: normal, 1: pasty, 2: semi-liquid, 3: liquid) where scores ≥ 2 indicated diarrhea [15].

### 2.5. Detection of Arg HRV Shedding by Rotavirus Antigen ELISA and CCIF

RS were used for the detection of HRV antigen by ELISA and infectious viral particles by cell culture immunofluorescence (CCIF) as described previously [16]. RS were collected daily and processed by swirling 10 times in 8 mL of Diluent #5 then centrifuged 1026× *g* for 15 min at 4 °C. The supernatant was collected, treated with Gentamycin (100 µg/mL), aliquoted in 400 µL volumes, and stored at −20 °C until use.

For the ELISA, 96 well plates were coated with 400 ng/100 µL of 2KA4 bivalent nanobody, specific for group A RV VP6, in carbonate buffer pH 9.6, incubated overnight at 4 °C, washed twice with PBS Tween 0.05% (PBST), blocked with 300 µL/well of PBS pH 7.4 with 5% nonfat dry milk, incubated for 1 h at 37 °C, and then washed 3 times with PBST [16]. 100 µL of RS samples were then added to the plate in duplicate. Semi-purified attenuated Wa HRV (AttHRV) antigen or RS samples from HRV-negative Gn pigs were used as positive and negative controls, respectively. Plates were incubated for 1 h at 37 °C and then washed three times with PBST. 100 µL/well of nanobody 2KD1-HRP-conjugated diluted 1:4000 in PBST was added to the plate and incubated for 1 h at 37 °C [16]. Plates were washed 3 times with PBST. The plates were developed with ABST peroxidase substrate solution (Seracare, 5120-0033 [50-62-01], Milford, MA, USA) for 15–30 min at room temperature before being stopped with ABST stop solution (Seracare, 5150-0017 [50-85-01], Milford, MA, USA) diluted 1:5. The optical density (OD) was measured at 405 nm on a Glomax Discover microplate reader (Promega, Madison, WI, USA).

For CCIF, 100 µL/well of EMEM (MEM, 1% pen-strep, 1% HEPES) was added to 96 well plates containing MA104 cells (ATCC, Manassas, VA, USA) at 100% confluency and incubated for 2 h at 37 °C with 5% CO_2_. 50 µL of RS samples were then added to the plate in duplicate. AttHRV and EMEM served as positive and negative controls, respectively. Plates were centrifuged at 930× *g* for 1 h at 21 °C and then 50 µL of EMEM containing trypsin (0.5 µg/mL, Sigma, cat#T-0303, St. Louis, MO, USA) was added to each well. Plates were incubated for 18 h at 37 °C with CO_2_. Plates were fixed with 100 µL/well of 80% acetone for 15 min, air dried for 45 min, and stored at −20 °C until staining. Plates were rehydrated with 300 µL/well of PBS for 5 min. Then, 40 µL/well of a broadly reactive anti-group A rotavirus nanobody 2KD1 conjugated with Alexa 488, diluted 1:500 in PBST was added and plates were incubated for 1 h at 37 °C [16]. Following washing 3 times with PBST, plates were observed with a fluorescent microscope and the total number of infectious particles was recorded.

### 2.6. Statistical Analysis

Gn pigs were randomly assigned into treatment groups, regardless of sex and body weight, after derivation by animal care technicians. Kruskal–Wallis test followed by Dunn’s test for multiple comparisons was used to analyze cumulative diarrhea scores, days with diarrhea, area under the curve (AUC) of diarrhea, shedding onset day by ELISA and CCIF, and days with shedding by ELISA and CCIF. Ordinary one-way analysis of variance (ANOVA) was used to analyze AUC of CCIF titers, mean peak titers, and mean duration of diarrhea. Two-way ANOVA with repeated measures followed by Tukey’s test for multiple comparisons was used to analyze CCIF titers and antigen ELISA OD values. Friedman’s test was used to analyze diarrhea scores of individual pigs. The correlation between inoculum dose, virus shedding, and diarrhea parameters was evaluated by Spearman’s test. All analyses were carried out using GraphPad Prism 8.0 (GraphPad Software, San Diego, CA, USA). A p value lower than 0.05 was accepted as statistically significant.

## 3. Results

### 3.1. Next Generation Sequencing

A total number of 320,379,476 reads from RNA and 106,873,069 reads from DNA were taken. Of these, the total number of reads for group A rotavirus gene segments is 181,816, accounting for 0.06% of the total number of reads from RNA. The vast majority were mapped to segment 11 (161,997), followed by segment 8 (15,531). The constellation of the Arg HRV strain (Arg 12461) was confirmed to be G4P[6]-I1-R1-C1-M1-A8-N1-T7-E1-H1. The complete nucleotide sequences of the 11 gene segments have been deposited to GenBank ((OP971570-OP971580). The only virus detected at a high copy number was group A HRV which was expected. All other contaminating DNA and RNA viruses detected in the sample were at extremely low copy numbers and without porcine pathogens.

### 3.2. Passaging of Arg HRV in Neonatal Gn Pigs

Twenty-two neonatal Gn pigs in total were used for serial passaging of Arg HRV stock inoculum and generating the inoculum pool. All pigs inoculated with Arg HRV, regardless of the passage number, developed diarrhea and shed the virus in feces. The mean days with diarrhea ranged from 2 to 3 and the mean cumulative score was 6–8. The majority of pigs shed virus from PID 1 to 2 or 3. The AUC of virus shedding remained consistent until passage 4, after which it began increasing and peaked at the final passage (Figure 1). However, this was not a statistically significant increase. The same pattern is observed in the detection of viral titers in intestinal contents, as titers began to rise at passage 4 and peaked at passage 6 (Table 1). This indicates the virulence of the Arg HRV strain was unchanged across passages. After passage 6 through Gn pigs, an inoculum pool with a titer of 3.4 × 10^5^ FFU/mL was generated.

### 3.3. Morphological Changes in Intestines of Arg HRV-Infected Gn Pigs

Animals infected with Arg HRV developed classical lesions seen in HRV infection in human infants (15–17). Lesions in the jejunum were less frequent and less severe than those in the ileum. There was evidence of sloughing apical villi in the jejunum. Injured jejunal villi had fibrin thrombi present (Figure 2a,b) and were associated with hemorrhage in nearby lamina propria (Figure 2c). Lamina propria in the jejunum were also affected by mild edema. A section of the lymph node that was incidentally captured during jejunum tissue collection was reactive with a diffuse expansion of lymphoid follicles, as well as hyperplasia of germinal centers and increased numbers of lymphocytes and plasma cells in the medulla.

The ileum of infected pigs suffered more extensive damage, including hydropic degeneration of enterocytes (Figure 3a) and diffuse villous blunting and atrophy (Figure 3a,c–g) consistent with lamina propria damage (Figure 3d). Peyer’s Patches and other lymphoid tissue in the ileum were hyperplastic with the expansion of germinal centers, consistent with active infection. The duodenum was insignificantly affected in all tissue samples examined.

### 3.4. Determination of an Optimal Challenge Dose of Arg HRV in Older Gn Pigs

The optimal challenge dose was defined as the dose of Arg HRV causing diarrhea and virus shedding in 100% of inoculated, naïve pigs at the age (33–34 days) when they will be challenged for evaluation of vaccine efficacy. A summary of clinical signs of infection is presented in Table 2.

At all dose levels (10^−2^, 10^3^, 10^4^, 10^5^), all pigs developed diarrhea and shed the virus in feces. In general, there was a positive association between the dosage and severity of clinical signs of infection despite being statistically insignificant by Spearman’s correlation test. Pigs in the 10^5^ group had diarrhea for longer than pigs in the lower dose groups (Figure 4A). The mean duration of days of diarrhea was 5.75 for the 10^5^ group versus 4.5, 5.25, and 3 for the 10^4^, 10^3^, and 10^−2^ groups, respectively. The mean cumulative fecal score and the AUC of diarrhea were also positively correlated with dosage (Figure 4B,C). Pigs in the 10^5^ group had 16 and 14.38 for the mean cumulative fecal score and AUC of diarrhea, respectively. In comparison, the remaining groups had mean cumulative fecal scores ranging from 10–14.25 and an AUC of diarrhea from 8.75–13.51. Mean days to onset of diarrhea were comparable across all groups (Figure 4D).

The mean duration of virus shedding was lowest for the 10^−2^ dose group at 2.5 days. The remaining three groups shed the virus for 5–5.75 days on average (Figure 5A). The mean peak titer and AUC of virus shedding were the greatest for pigs in the 10^5^ group and lowest for the 10^−2^ group (Figure 5B,C). Mean days to onset of virus shedding was statistically different between the 10^−2^ group and the two highest groups, but otherwise, onset was similar across other groups (Figure 5D). Daily CCIF titers were consistent across all dosage groups (Figure 6). The majority of dosage groups had resolved the infection and disease, as evidenced by the discontinuation of virus shedding, by PID 7. The only group to still be shedding virus on PID 7 was the 10^3^ group.

## 4. Discussion

The introduction of HRV vaccines has significantly decreased the burden of disease in the United States, however, HRV induced gastroenteritis is still a significant issue in many low- and middle-income countries (LMIC). This is due in part to variable vaccine efficacy in LMICs compared to high income countries (HIC) and upper middle-income countries (UMIC) [2]. Proposed hypotheses for lower efficacy include differences in the gut microbiome, host factors such as breast milk constituents and HBGA secretor antigens, and higher transmission rates of HRV in LMIC [2]. The emergence of new P[6] serotypes in LMIC also contributes to their higher rates of rotaviral disease and lower vaccine efficacy. Rotarix and RotaTeq efficacy is markedly lower against these strains than the predominant P[8] strains [4]. Additionally, many of the novel HRV strains have animal origins. Their chimeric nature gives them a unique chance to spread more efficiently as the possession of genes from multiple hosts eases the pressure of adaptation in new environments [10]. In LMIC where living space may be shared with livestock, the risk of zoonotic transmission rises [5]. P[6] HRVs have been isolated from human fecal samples since the 1990s [17,18]. Their continued detection in addition to their zoonotic potential highlights the need for broadly protective vaccines and an appropriate animal model to test said vaccines.

Gn pig models of infection and disease are extremely useful tools that can address this need. While a Gn pig model of Wa P[8] HRV is well-established, this is to our knowledge the first Gn pig model of P[6] HRV infection and disease. The successful propagation and continued development of clinical signs of infection indicated the infectivity and pathogenesis of the virus in neonatal Gn pigs. The reliable infectivity of Arg HRV in Gn pigs is likely influenced by its porcine origins. It is probable that this strain is the result of interspecies recombination due to its matching P and G type with the porcine Gottfried strain (G4P[6]). Additionally, ten genes of Arg HRV are of porcine origin with six being closely related to the Gottfried strain [5]. This hypothesis is further evidenced by the lack of attenuation during passaging of Arg HRV in neonatal Gn pigs. The viral titer of pooled intestinal contents remained consistent across passages, with the lowest measured being 2 × 10^4^ FFU/mL. Intestinal lesions localized to the jejunum and ileum, with the ileum suffering more damage, indicated significant pathogenicity. The development of clinical signs of infection in these animals is similar to those seen in Gn pigs infected with the Gottfried strain. Neonatal and older Gn pigs are susceptible to infection with the Gottfried strain, with both age groups developing diarrhea and shedding the virus in feces [19]. The onset of diarrhea in neonatal animals ranged from PID 1–3 for those infected with Gottfried and from PID 1–2 for those infected with Arg HRV [19]. Neonatal Gottfried infected pigs had diarrhea and virus shedding for 5.5 days [19]. Determining the duration of diarrhea and virus shedding for Arg HRV infected neonatal pigs was not feasible in this study due to the need to collect intestinal contents for serial passaging at the moment of peak virus replication. In older animals, Gottfried infection induced virus shedding for 5 days on average [19]. The outcomes from our dose response study support these results, as we found that pigs in the 3 highest dose (10^3^, 10^4^, and 10^5^) groups shed the virus in feces for approximately 5 days on average (5.75, 5.25, and 5 days, respectively).

Arg HRV infection in Gn pigs also has great similarities to virulent Wa HRV strain infection. A study from Ward et al. [13] showed that Gn pigs are highly susceptible to Wa HRV infection, even at doses as low as <1 FFU [13]. Following both Arg and Wa HRV infection, 100% of pigs shed the virus in feces regardless of the dose [13]. The mean onset day and mean duration were similar between the two models, ranging from PID 1–2.5 or PID 1.3–2 for Arg HRV and Wa HRV, respectively [13]. Pigs infected with the lowest dose of Arg HRV had a delayed onset of virus shedding, and this was also true for pigs infected with Wa HRV. However, Wa HRV infected pigs had higher mean peak titers compared to Arg HRV infected pigs. For example, pigs infected with <1 FFU of the Wa HRV strain had a mean peak titer of 1 × 10^6^ FFU/mL whereas dose matched Arg HRV infected animals had a mean peak titer of 5.4 × 10^3^ FFU/mL [13]. This trend was consistent across all doses. Furthermore, in Arg HRV infected pigs, the mean peak titer was positively correlated with dose, but this was not observed with the Wa HRV strain. When comparing the onset and severity of diarrhea, in general, there was a high degree of similarity between dose matched groups. The onset of diarrhea for Wa HRV infected pigs ranged from PID 1.5–3 and the onset of virus shedding ranged from PID 1.3–2 [13]. The Arg HRV inoculated pigs had the onset of diarrhea ranging from PID 1–4, with PID 1–2 being the most common onset day, and there was no statistically significant difference among dose groups. For the onset of virus shedding, it ranged from PID 1–3, with PID 1 being the most common onset day, and it did not differ among the three higher dose groups.

The development of intestinal lesions in neonatal Gn pigs infected with Arg HRV was similar to those seen in older Wa HRV infected animals. Lesions tended to cluster within the jejunum and ileum and included villous atrophy, reactive hyperplasia of Peyer’s patches, and accumulation of edema [13]. In all Arg HRV infected animals, the duodenum was insignificantly affected. Comparatively, ~60% of Wa HRV infected animals had duodenal lesions [13]. This is an interesting difference between the virus strains. It is unlikely that duodenal lesions were missed during specimen collection from Arg HRV infected animals as none had abnormal duodenal physiology. By PID 7, all animals had cleared infection as indicated by the resolution of diarrhea and returned to normal intestinal physiology [13]. No animals had signs of chronic infection. This pattern matches the trend seen in rotavirus infections in humans where immunocompetent hosts typically resolve gastrointestinal symptoms within 3–7 days [20].

Several factors must be taken into consideration when determining the optimal challenge dose for Gn pigs. First, the dosage should be high enough to induce clinical signs of disease in 100% of animals. For Arg HRV infection, these signs include the development of diarrhea and virus shedding in feces. In our dose–response study, 100% of pigs in all groups experienced the disease. This suggests that the development of diarrhea and virus shedding may be independent of dosage. In the Gn pig model of virulent Wa HRV, virus shedding and diarrhea development were also independent of dosage [13].

The severity of infection and disease, as measured by AUC, mean peak titer, and cumulative fecal scores also needs to be taken into consideration. For the optimal challenge dose, infection severity should be enough to recapitulate natural infection but not so severe as to overwhelm the immune system and skew the efficacy outcomes of potential therapeutics. While there were interesting trends regarding the association between the higher inoculum doses and increased disease severity, none were statistically significant. Across the board, the 10^−2^ dose group had the lowest mean cumulative fecal score, AUC of virus shedding, AUC of diarrhea, and mean peak titer. Comparatively, the 10^5^ dose group had the highest scores in all these categories. Some animals had a duration of diarrhea that was longer than virus shedding, which is expected. Rotavirus diarrhea at the very early stage of infection is triggered by NSP4, the enterotoxin produced by the virus. Later on, diarrhea is also contributed by malabsorption from villous atrophy caused by rotavirus infection [21]. In some animals, virus shedding is no longer detectable by CCIF before the recovery of the function of the enterocytes damaged by the virus.

In theory, a dose of 10^−2^ FFU would take longer to produce disease of the same severity as a 10^5^ dose. However, innate immune responses such as Type III interferons and natural killer T cells would likely inhibit infection before the disease became severe [22,23,24]. Because of this, it is reasonable to think the dose group receiving 10^−2^ FFU of inoculum would experience lower peak titers and AUC values as well as a shorter duration of both virus shedding and diarrhea. This was the pattern observed in our study where the 10^−2^ group had the lowest values of all groups across all criteria mentioned above. Furthermore, the 10^−2^ group had delayed onset of virus shedding at a statistically significant level. Therefore, it may be that disease severity is only independent of inoculum dose above a certain threshold. It has been demonstrated for other viruses, such as influenza, respiratory syncytial virus, and coronaviruses that dose (measured as viral load) may play a direct role in disease severity [25,26]. A future study with more nuanced dosage level differences would be necessary to elucidate this with certainty for Arg HRV.

## 5. Conclusions

Because of the similarities between the development of intestinal lesions, virus shedding, and diarrhea as seen in the Wa P[8] HRV Gn pig model, we have determined the optimal challenge dose of Arg HRV to be 10^5^ FFU in Gn pigs. This model is ready to serve the need for preclinical evaluations of candidate multivalent HRV vaccines against homotypic and heterotypic HRV infection and disease.

## Figures and Tables

**Figure 1 viruses-14-02803-f001:**
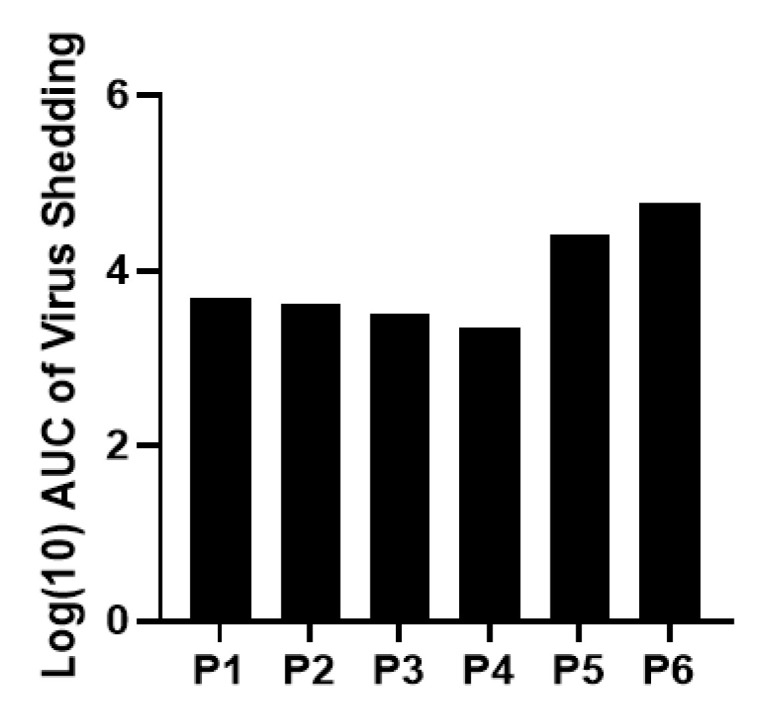
Arg HRV titers through passaging in neonatal Gn pigs. Data presented are log10 transformed area under the curve (AUC) of virus shedding in feces. Peak virus shedding occurred during passage 6.

**Figure 2 viruses-14-02803-f002:**
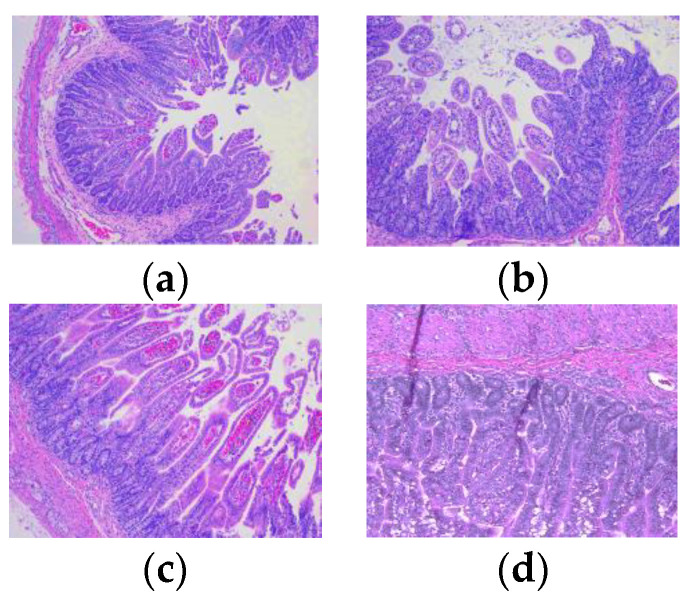
Hematoxylin and eosin-stained jejunal lesions in Gn pigs following oral inoculation with Arg P[6] HRV. All tissues were collected at necropsy on PID 2 or 3. (**a**) Villous damage with fibrin thrombi. (**b**) Atrophy of the enterocytes. (**c**) Hemorrhage is present in the lamina propria. (**d**) Normal jeju num morphology of a mock-infected Gn pig. The magnification of all images is 100×.

**Figure 3 viruses-14-02803-f003:**
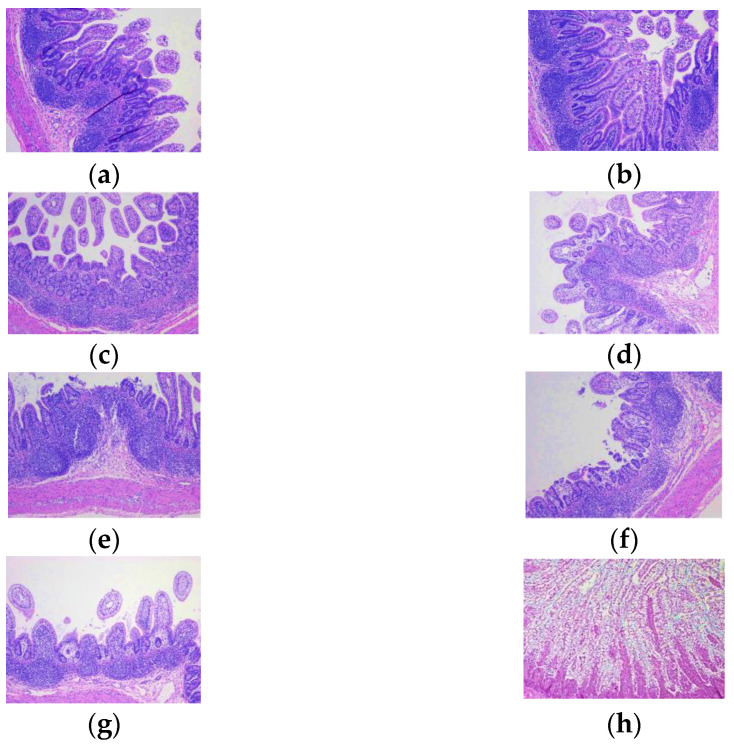
Hematoxylin and eosin-stained ileal lesions in Gn pigs following oral inoculation with Arg P[6] HRV. All tissues were collected at necropsy PID 2 or 3. (**a**–**g**) or mock (**h**). Villous blunting/atrophy (**a**,**c**–**g**), hydropic degeneration (**a**), and lymphoid hyperplasia (**a**,**b**,**d**,**e**,**g**) are present. Edema is also present in the lamina propria (**d**). (**h**) Normal ileum morphology of a mock-infected Gn pig. The magnification of images (**a**–**g**) is 100×, and the magnification for image H is 200×.

**Figure 4 viruses-14-02803-f004:**
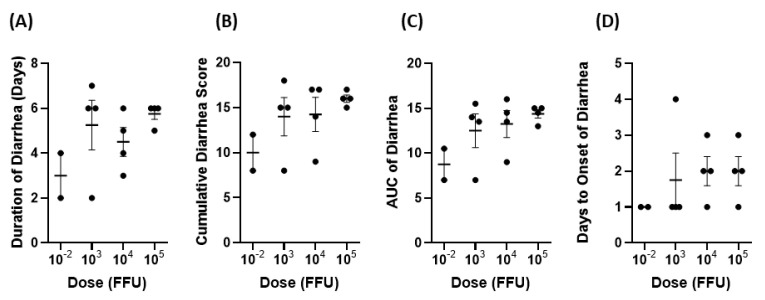
Diarrhea from PID 1–7 after inoculation of Gn pigs with different doses of Arg HRV. (**A**) mean duration of diarrhea, (**B**) mean cumulative diarrhea scores, (**C**) AUC of diarrhea, and (**D**) onset of diarrhea. Error bars represent SEM.

**Figure 5 viruses-14-02803-f005:**
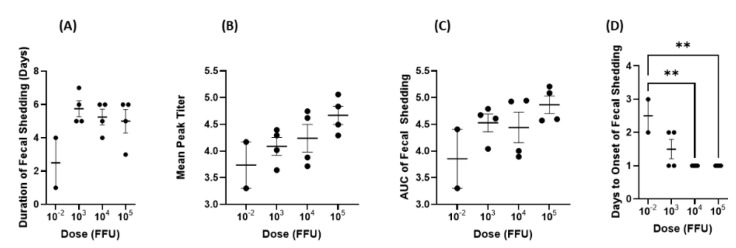
Virus shedding from PID 1–7 after inoculation of Gn pigs with different doses of Arg HRV. (**A**) duration of fecal shedding, (**B**) geometric mean peak titer, (**C**) AUC of fecal shedding, and (**D**) onset of fecal shedding. A log(10) transformation of data was performed for the geometric mean peak titer and AUC before graphing. Geometric mean peak titers and AUC of fecal shedding were positively correlated with inoculum doses (r = 1, *p* = 0.08). Error bars represent SEM. ** *p* < 0.01.

**Figure 6 viruses-14-02803-f006:**
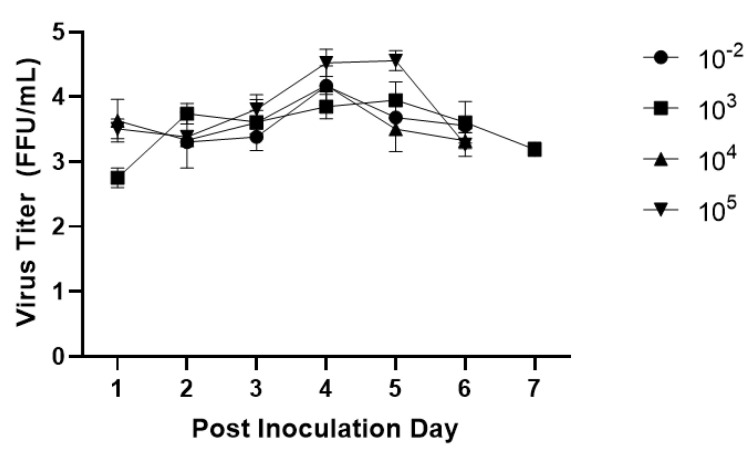
Mean daily virus shedding titer of Gn pigs with different doses of Arg HRV. Log(10) transformed data are presented. All dosage groups shed approximately the same level of the virus as determined by CCIF.

**Table 1 viruses-14-02803-t001:** Viral titers in intestinal contents collected after passaging. Small and large intestinal contents were collected from Gn pigs for each passage and were pooled. The viral titers were determined by CCIF. P0 refers to the initial titer of the infant stool sample before any passaging in pigs.

n/Passage	Passage Number	Titer (FFU/mL)
-	P0	5.6 × 10^6^
1	P1	2 × 10^4^
2	P2	Not determined
3	P3	4 × 10^4^
2	P4	7.2 × 10^4^
6	P5	1.67 × 10^5^
5	P6	5.7 × 10^6^

**Table 2 viruses-14-02803-t002:** HRV fecal shedding and diarrhea after inoculation of Gn pigs with different doses of Arg HRV (G4P[6]).

Diarrhea							Virus Shedding				
Dose Group	Inoculum Dose (FFU/mL)	n	Percentage with Diarrhea	Mean Days to Onset	Mean Duration (Days)	AUC of Diarrhea	Percentage Shedding Virus	Mean Days to Onset	Mean Duration (Days)	Mean Peak Titer (FFU/g of Feces)	AUC of Virus Shedding
1	10^−2^	2	100	1.0	3.0	8.75	100	2.5	2.5	5441	13,800
2	10^3^	4	100	1.75	5.25	12.5	100	1.5	5.75	12,212	40,150
3	10^4^	4	100	2.0	4.5	13.5	100	1.0	5.25	17,388	47,650
4	10^5^	4	100	2.0	5.75	14.38	100	1.0	5.0	46,771	90,200

Note: Gn pigs were orally inoculated with Arg HRV G4P[6] at 33–34 days of age. Rectal swabs were collected daily after inoculation from postchallenge day (PCD) 0–7 to evaluate virus shedding. Rotavirus shedding titers were determined by rotavirus antigen ELISA (detect viral antigen) and CCIF (determine the number of infectious viral particles). Fecal consistency scores were used to assess diarrhea from PCD 0–7; scores are defined as 0: solid, 1: pasty, 2: semi-liquid, and 3: liquid. Scores of 2 or higher are considered diarrheic. AUC, area under the curve.

## Data Availability

All relevant data are included within the manuscript.

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
