# Peer review of "A New Gnotobiotic Pig Model of P[6] Human Rotavirus Infection and Disease for Preclinical Evaluation of Rotavirus Vaccines"

_viruses, 2022, doi:10.3390/v14122803_

Round 1

Reviewer 1 Report

Testing vaccines in animal models is a critical step in vaccine development.

The article describes an animal model for pre-clinical assays of a potential rotavirus vaccine that contemplate the VP4 genotype P[6].

Authors highlight that the most widely used vaccines are efficient against P[8] strains but are much less successful  against P[6] strains of rotavirus that have been emerging across the globe.

Another aspect remarked is that pigs are highly susceptible to many rotavirus genogroups, including the human-predominant Group A HRVs. Furthermore, genomic analysis of the G4P[6] strains suggested that they are of porcine origin.

The consistencies in clinical presentation to human children in addition to similar immune system physiology and development make Gn pigs invaluable for modeling HRV infection and disease.

All of the above suggests that pigs ‘are an appropriate candidate for testing rotavirus P[6] vaccines.

Criteria evaluated in this work were as listed below

- viral titers in intestinal contents collected after passaging

- developing of diarrhea and virus shedding in feces

- duration of diarrhea and shedding

- Morphological changes in intestines

ID50 and DD50 were used to select the optimal dose for challenge

Methods

Inoculum was generated in Gn pigs inoculated with 5.6 x 106 (FFU)/mL of Arg G4P[6] HRV and intestinal contents were collected and used to continue serially passaging of the virus in Gn pigs. The presence of other RNA or DNA viruses was discarded by Next generation sequencing.

Histopathological changes were analyzed from Gn pigs inoculated with 105 ffu of Arg HRV from the 6th passage through Gn pigs.

Results

AUC of shedding and after began increasing after passage 4, and peaked at the final passage, not statistically significant. Same pattern was observed in titers of intestinal content, indicating virulence was unchanged across passages.

Table 1: It is correct that P1 titer is 106?

Table 4:

Please check the values of mean peak titer and AUC of virus shedding, also the metric units for all the variables must be in the table.

I suggest clarifying table 4, data are difficult to read and understand. It could be simplified (most of data in the table are shown in figures 4 and 5). One suggestion is to omit the letter A and B that indicates the significance of the different values (also considering that SEM are shown in figures 4 and 5 and significances are mentioned in the text). A division between diarrhea and shedding within the table itself could help. Percentages in parentheses are not necessary for this n and values.

The notes at foot table are enumerated as a, b, c, etc., but this ordering does not match with the content of the table.

What is the discussion for groups with:

-             less time for diarrhea onset with lower doses?

-             duration of diarrhea longer than shedding?

Please note that values for 102 o in Figure 6 do not correspond with values in figure 5A.

In this work, assays showed successfully that a rotavirus strain with an emerging VP4 P[6], can replicate and cause clinical signs of infection in Gn pigs. These are critical steps for animal models for pre-clinical studies. The low number of individuals in each group weakens statistical analysis, however, the results provide sufficient relevant data for the first steps in the development of an animal model for rotavirus unusual genotypes.

It is important to notice that there are some inconsistencies between results that need to be addressed in the discussion. Also, there are discrepancies between important data in different analyses (that were pointed out). These aspects need to be corrected before considering the article to be published.

Reviewer 2 Report

Nyblade et al report a new gnotobiotic (Gn) pig model of P[6] HRV infection.Gn pig developed clinical signs of Arg HRV infection. The optimal challenge dose was determined evaluating potential vaccine candidates.There are still some shortcomings that can be improved to improve the readability of this study.

1. Line 177-180 only gives the total number of reads of the Next generation sequencing, at least how many reads are mapped to the virus, especially the number and proportion of reads in group A HRV?

2,Line 182-183 in the process of Passaging of Arg HRV and generating the inoculum pool in neonatal Gn pigs, should the whole intestinal contents or part of the diseased intestine (jejunum, ileum, etc.) be collected?

3. It is mentioned in the paper that Gn pig model of HRV infection needs to determine the median infectious dose (ID50). median diarrhea dose 63 (DD50), however, only the optimal challenge dose of Arg HRV (105 FFU) in Gn pigs was given in this paper.What were ID50 and DD50 of Arg HRV?

4,There were mistakes in Table 1,such as missing in the Passage column 4. Why is the virus titer of P3 NA?

5. The pathological maps in Figure 2 and Figure 3 did not show the magnification factor. Please add the magnification factor or scale.
